# Visual Tract Degradation in Bilateral Normal-Tension Glaucoma—Cortical Thickness Maps and Volumetric Study of Visual Pathway Areas

**DOI:** 10.3390/jcm11071907

**Published:** 2022-03-29

**Authors:** Anna Pankowska, Sylwester Matwiejczuk, Paulina Kozioł, Tomasz Żarnowski, Radosław Pietura, Ewa Kosior-Jarecka

**Affiliations:** 1Department of Radiography, Medical University of Lublin, 20-079 Lublin, Poland; paulina.koziol@op.pl (P.K.); radoslawpietura@gmail.com (R.P.); 2Department of Diagnostics and Microsurgery of Glaucoma, Medical University of Lublin, 20-079 Lublin, Poland; sylwester.matwiejczuk@gmail.com (S.M.); zarnowskit@poczta.onet.pl (T.Ż.); ewakosiorjarecka@umlub.pl (E.K.-J.)

**Keywords:** glaucoma, NTG, MRI, lateral geniculate nucleus, cortical thickness

## Abstract

The aim of the study was to evaluate changes in the central visual pathways during the early and advanced stages of bilateral normal-tension glaucoma (NTG). Methods: The studied groups constituted patients with bilateral normal-tension glaucoma of the same stage (*n* = 45) and age-matched healthy volunteers (*n* = 17). All patients underwent ophthalmic examination and examination on a 1.5 Tesla Magnetic Resonance Scanner (Optima 360, GE Healthcare). Volume and cortical thickness analyses were performed using the open-source automated software package FreeSurfer. Results: There was a significant difference in lateral geniculate nuclei volume between the control and advanced glaucoma groups in the right hemisphere (*p* = 0.03) and in the left hemisphere between the early and advanced glaucoma patients (*p* = 0.026). The optic chiasm volume differed significantly between the control and advanced NTG groups (*p* = 0.0003) and between early and advanced glaucoma patients (*p* = 0.004). Mean cortical thickness analysis revealed a significant increase in values in the advanced glaucoma group in the right Brodmann area 17 (BA17) (*p* = 0.007) and right BA18 (*p* = 0.049) as compared to early NTG. In the left BA18 area, the mean thickness of the cortex in the early glaucoma group was significantly lower than in the control group (*p* = 0.03). Conclusions: The increase in the grey matter thickness in the V1 region with more-advanced glaucoma stages may reflect compensatory hypertrophy. Additionally, the regions of the brain early affected during glaucoma with reduced thickness were the right lateral occipital gyrus and left lingual gyrus. The most prominent change during the course of glaucoma was the increase in grey matter thickness in the right cuneus.

## 1. Introduction

Glaucoma is a heterogeneous group of diseases caused by the death of retinal ganglion cells (RGCs), and clinically characterized by the excavation of the optic nerve head and visual-field loss [1,2]. Glaucoma is the second-leading cause of blindness worldwide and the most frequent cause of irreversible blindness worldwide [2,3]. In Europe, the prevalence of glaucoma is 2.93% among persons aged 40 to 80 years. The prevalence rises with age, reaching 10% in persons over 90 years old [4].

The causes of RGC death in glaucoma remain unclear. Increased intraocular pressure (IOP) remains the most important and only modifiable risk factor for glaucoma development. However, there is an enormous group of patients who develop glaucoma, so-called normal-tension glaucoma (NTG), in the absence of IOP elevation. In the case of NTG, risk factors different from IOP elevations are suggested. Among them are a higher sensitivity to normal pressure, vascular dysregulation, an abnormally high translaminar pressure gradient, and a neurodegenerative process due to impaired cerebrospinal fluid dynamics in the optic nerve sheath compartment [5].

Generally, two principal theories for the pathogenesis of glaucomatous optic neuropathy have been described as a mechanical and a vascular theory [6]. Moreover, the studies suggest that glaucoma is similar to neurodegenerative diseases such as Alzheimer’s disease and Parkinson’s disease [7,8], with neurodegenerative changes localized in the central nervous system. The latter association is also supported by studies showing trans-synaptic degeneration affecting the central areas of the visual system in patients with glaucoma [7]. This shed new light on the disease, giving life to the new theory suggesting that the disease could be considered not only an ocular disease but a more complex neurodegenerative process that affects the entire visual system [7]. However, the involvement of the central nervous system during the course of glaucoma remains unclear, including whether the observed pathology is the cause or rather the result of the ongoing glaucomatous process.

The studies of the central involvement during glaucoma pathology also give another interesting possibility. Primary open-angle glaucoma (POAG) is painless and asymptomatic for a long time. Visible changes at the head of the optic nerve and defects in the visual field are detected when 40% of RGC is irreversibly lost [9]. For this reason, there is a clear need for the methods enabling the introduction of earlier detection. One of the promising techniques is magnetic resonance imaging (MRI) [10].

Normal-tension glaucoma (NTG), a subtype of POAG, occurs despite the IOP never exceeding the normal range. Recent studies have revealed that IOP-independent risk factors, including vascular factors, trans-laminar pressure gradient, immune-related disorders, genetic factors, or disturbances of the biomechanical properties of the eye, may play a crucial role in NTG pathogenesis [11,12]. Additionally, the relationship between glaucoma and cognitive impairment, especially that due to Alzheimer’s disease and Parkinson’s disease, was reported [13,14,15]. Most MRI studies did not separate NTG patients from the whole POAG group. However, patients with NTG were described to have abnormalities in the anterior visual pathway involving the optic nerve diameter, the height of the optic chiasm, and LGN volume [16]. Studies with the diffusion tensor imaging (DTI) technique indicated whole-brain white matter damage in NTG patients and showed a positive correlation with RNFL thickness [17,18].

In this study, we have applied both voxel-based morphometry (VBM) and surface-based registration (SBR) methods for investigating anatomical alterations that occur in the brain during the glaucomatous process on a different degree of advancement. We measured volumes of the lateral geniculate nuclei (LGNs) and optic chiasm and cortical thickness in areas associated with vision using an open-source neuroimaging toolkit FreeSurfer (Massachusetts General Hospital, Harvard Medical School; http://surfer.nmr.mgh.harvard.edu, 27 January 2022).

The study aimed to evaluate the changes observed in the central visual pathways during the early and advanced stages of NTG.

## 2. Materials and Methods

The study was designed and conducted in the Department of Diagnosis and Microsurgery of Glaucoma and Department of Radiography in the Medical University of Lublin (Poland) with the acceptance of the Local Ethics Committee (approval number KE 0254/149/2018), and performed in accordance with the standards set by the Declaration of Helsinki. Before the study, all participants signed informed consent and underwent ophthalmic examination.

### 2.1. Participants

The studied groups contained patients with bilateral normal-tension glaucoma of the same stage (*n* = 45) and age-matched healthy volunteers (*n* = 17).

Patients were diagnosed with NTG according to the following criteria: glaucomatous neuroretinal rim loss, open angle in gonioscopy, and intraocular pressure (IOP) consistently <21 mm Hg (with the highest value ever measured in a given patient being ≤21 mm Hg, maximal IOP was evaluated by 4 measurements during office-hours), and no detectable eye pathology which would imply a diagnosis of secondary glaucoma. During the diagnostics, the maximal IOP was checked during 2 separate visits during different time points and, during every visit, it was measured twice with a 2 h break. Diagnosis was established by experienced ophthalmologists (EKJ, TZ).

The Humphrey visual field test 24-2 was performed on all patients The VF results were evaluated and the diagnosis was confirmed by glaucoma specialists (EKJ). Glaucomatous defect on SAP was defined based upon a glaucoma hemifield test result outside normal limits and the presence of at least 3 contiguous test points within the same hemifield on the pattern deviation plot at *p* < 1%, with at least 1 point at *p* < 0.5%, on at least 2 consecutive tests, with reliability indices better than 15%. According to the criteria of Hodapp–Parrish–Anderson in visual field examination, patients were divided into two subgroups: early glaucoma (MD better than −6 dB; *n* = 28) and advanced glaucoma (MD less than −12 dB; *n* = 17). Only the patients with bilaterally the same stage of the defect were included.

The control group was recruited from the patients who, after ophthalmic diagnostics, had excluded glaucoma and other diseases possibly influencing visual pathways (retinal or optic nerve disorders, central nervous system disorders). Demographic and clinical data of studied groups are presented in Table 1.

### 2.2. MRI Data Acquisition

All patients underwent examination on a 1.5 Tesla Magnetic Resonance Scanner (Optima 360, GE Healthcare) using a GE HD 8 channel neurovascular array coil. Standard brain and orbits protocol was extended by a 3D T1-weighted magnetization-prepared rapid acquisition gradient echo (MPRAGE) sequence acquired with the following parameters: TR = 9.4 ms, TE = 3.9 ms, FA = 8, TI = 1000 ms, FOV = 24 cm, matrix 192 × 192, slices = 166, voxel size 1.2 × 1.2 × 1.2 mm^3^. During 6 min acquisition, the patients were asked to remain still and to avoid any eye movement. MRI data acquisition was performed by experienced radiologists (RP, AP, PK).

### 2.3. Data Analysis

Volume and cortical thickness analyses were performed using the open-source automated software package FreeSurfer (version 7.1.1, Massachusetts General Hospital, Harvard Medical School; http://surfer.nmr.mgh.harvard.edu, 27 January 2022). Analyzed data were acquired using an automatic recon-all procedure in FreeSurfer containing the following steps: motion correction and intensity normalization, Talairach transformation, skull stripping and neck removing, registration, anatomical structure labeling, and white matter segmentation based on intensity, neighborhood, and smoothness constraints. Next, deformable surfaces models were created, starting from the tessellation process. Before generating final surfaces (white, pial, inflated) and registration to spherical atlas, automated topology fixer was used. Cortical thickness in that approach was calculated as the distance between white and pial surfaces.

Each subject underwent a quality assessment performed by an experienced radiologist. Prepossessing steps including brain mask and skull stripping check, normalization, and segmentation evaluation were executed manually to excluding the presence of topology defects and automated pipeline errors.

The volume of LGNs was measured in selected groups using a thalamic nuclei atlas developed by Iglesias et al. [19]. Optic chiasm volume assessment was performed according to data obtained from the Desikan and Killiany parcellation atlas [20].

The current study focuses on cortical visual areas selected from different anatomical atlases which may be affected by the glaucomatous process. We have selected 3 areas from Brodmann’s area atlas (V1–BA17, V2–BA18, V5/MT–BA19) [21], and 5 areas from Destrieux atlas (lingual gyrus, calcarine sulcus, cuneus, collateral sulcus and lingual sulcus, occipital pole) [22,23], and calculate mean cortical thickness for each hemisphere.

Moreover, statistical cortical thickness maps were calculated using FreeSurfer’s Query, Design, Estimate, Contrast (QDEC) graphical user interface. After smoothing cortical thickness measures using a full width at half maximum (FWHM) Gaussian kernel of 10 mm, each group and corresponding hemispheres were compared with each other using a General Linear Model (GLM) computed vertex-by-vertex. Results were presented with and without correction for multiple comparisons.

### 2.4. Statistical Analysis

PQStat software package ver. 1.8.2 was used for the statistical analysis of the mean volume and mean cortical thickness results. For group comparison, one-way analysis of variance (ANOVA) or Welch’s ANOVA (in the case of unequal variances) was employed. For multiple comparison procedures, parametric Tukey’s (when equal variances) and nonparametric Games–Howell’s (when variances were unequal) post hoc tests were performed. When rejecting outliers, Chauvenet’s criterion was applied. The results were considered significant at *p* < 0.05.

Significant results of cortical thickness change calculated using QDEC after clusterwise correction for multiple comparisons with a Monte Carlo Simulation (*p*-value threshold of <0.01) were presented on semi-inflated cortical surfaces. Results presented on maps were considered significant at clusterwise probability (CWP) < 0.05.

## 3. Results

Data obtained during thalamic parcellation and subcortical segmentation showed that there is a significant difference in lateral geniculate nuclei (LGNs) volume between the control (238.5 mm^3^) and advanced glaucoma groups (208.2 mm^3^) in the right hemisphere (*p* = 0.03), and in the left hemisphere between the early (240.5 mm^3^) and advanced glaucoma (217.0 mm^3^) patients (*p* = 0.026). Furthermore, optic chiasm volume differs significantly between the control (225.3 mm^3^) and advanced glaucoma (156.9 mm^3^) groups (*p* = 0.0003) and between early (206.2 mm^3^) and advanced NTG patients (*p* = 0.004) (Figure 1).

Mean cortical thickness analysis in anatomical regions selected from Brodmann’s areas atlas revealed a significant increase in values in the advanced NTG group in the right BA17 (V1 area, *p* = 0.007) and right BA18 (V2 area, *p* = 0.049) as compared to early NTG. Moreover, in the left BA18 area, the mean thickness of the cortex in the early glaucoma group was significantly lower than in the control group (*p* = 0.03) (Figure 2).

In the second analyzed anatomical atlas (Destrieux), the cortex was significantly thinned in the early stage of glaucoma as compared to the advanced stage in the cuneus (right, *p* = 0.013; bilateral, *p* = 0.034), left collateral sulcus and lingual sulcus (*p* = 0.034), and in the left occipital pole (*p* = 0.013). A significant increase in mean cortex thickness was observed in advanced NTG in comparison to the control group in the left occipital pole (*p* = 0.014). All results are presented in Table 2.

FreeSurfer’s analysis performed in QDEC revealed visual regions from the Desikan and Killiany anatomical atlas of altered cortical thickness. Thinning of the cortex was observed in the early and advanced glaucoma groups in the right lateral occipital gyrus and in the left lingual gyrus as compared to the control group. Furthermore, there was an increase in cortical thickness values in the right pericalcarine of the advanced NTG group (compared to control). Evaluation of cortical thickness between both NTG groups showed an increase in values in the early stage in the right lateral occipital gyrus and left lateral occipital gyrus and a decrease in the right cuneus, right lingual gyrus, and right lateral occipital gyrus. Results are presented in Table 3.

Results after correction for multiple comparisons revealed significant thinning in the early glaucoma group as compared to advanced glaucoma only in the right cuneus (CWP = 0.045), in a cluster area of 431.1 mm^2^. Results are presented in cortical statistical maps in Figure 3.

Correlation analysis, presented in Table 4, revealed positive dependency between average temporal RNFL and mean LGN volume in a parametric test (*r* = 0.34, *p* = 0.027) in the early NTG group, and negative dependency between average nasal RNFL and mean LGN volume in a nonparametric test (*R* = −0.50, *p* = 0.035) in the advanced NTG group (compared bilaterally with corresponding eyes). When comparing opposite eyes, mean LGN volume correlated positively in the early glaucoma group with average RNFL (*r* = 0.38, *p* = 0.012; *R* = 0.46, *p* = 0.002), average superior RNFL (*R* = 0.43, *p* = 0.005), average temporal RNFL (*r* = 0.39, *p* = 0.011; *R* = 0.50, *p* = 0.001), and average inferior RNFL (*r* = 0.37, *p* = 0.016; *R* = 0.36, *p* = 0.019), and negatively with average nasal RNFL in the advanced glaucoma group (*r* = −0.48, *p* = 0.043; *R* = −0.51, *p* = 0.029).

Table 5 shows the parametric (Pearson’s) and nonparametric (Spearman’s) correlations analyses based on clinical data (MD and average RNFL) and mean cortical thickness calculated in selected areas using FreeSurfer software. MD correlated negatively with V1 thickness in advanced NTG stage when comparing corresponding (*r* = −0.50, *p* = 0.009; *R* = −0.52, *p* = 0.00) as well as opposite sides (*r* = −0.40, *p* = 0.046; *R* = −0.48, *p* = 0.006). Furthermore, in the early NTG stage, correlations between MD and mean cortical thickness in the calcarine sulcus (*R* = 0.33, *p* = 0.034) and occipital pole (*r* = −0.30, *p* = 0.048; *R* = −0.35, *p* = 0.019) were noticed. There were no significant correlations observed between cortical thickness and average RNFL in any of the studied groups. Detailed results are presented in Table 5.

## 4. Discussion

Glaucoma is a disease with complex pathogenesis in which the damage is detectable along the whole visual pathway. Degenerative changes were reported in the pregeniculate part, including the optic nerves and optic tracts, and in postgeniculate pathways with the optic radiations and visual cortex [24,25,26]. Additionally, recent studies have reported degeneration in non-vision-related areas of the brain, suggesting the presence of a visual-pathway-independent brain involvement [27]. The two hypotheses explaining pathogenesis of the central involvement in glaucoma were proposed. One claims that the RGC loss leads to the transmission of the injury to the LGN and further to the visual cortex. However, there is also scant evidence showing that the LGN may be the primary site of the injury, with the secondary reduction in the axonal transport of neurotrophic factors causing RGC apoptosis [26,28]. The latter is proposed especially for NTG, where elevated IOP is not the main causative factor.

In its pathway to the brain, the optic nerve crosses the optic chiasm, so the LGN and primary visual cortex receive information from both the nasal visual field of the ipsilateral eye and the temporal visual field of the contralateral eye. This makes the assessment of the central visual pathway more difficult because the participation of the particular optic nerve in the central damage is unable to be estimated. To avoid this uncertainty, we decided to include only the patients with symmetrical binocular glaucomatous defects.

Changes in the optic chiasm are known in glaucoma. Zhang et al. examined the peripheral part of the visual pathway in NTG with MRI, and determined that constriction of the optic nerve and reduction in the size of the chiasm correlated with a decrease in RNFL [16]. Additionally, the decrease in the intracranial part of the optic nerve in glaucoma correlated with the histological decline in the number of RGC [29]. In this study, a significant decrease in optic chiasm volume was observed when glaucoma stages become more advanced. This difference reflects the proper inclusion of patients in the early and advanced glaucoma groups.

The LGN is the relay center in the thalamus for sensory information to enter the visual cortex. Furthermore, it plays a role in perception and cognition far beyond that of a relay nucleus, and it has to be considered as an early gatekeeper in the control of visual attention and awareness [30]. The LGN consists of six distinctive layers. The inner two layers are magnocellular layers, while the outer four layers are parvocellular layers. In between these two layers are koniocellular cells. Numerous studies using MRI imaging show changes in LGN in the course of glaucoma [31,32,33,34]. Similar changes were observed in animal models of glaucoma [35]. Our study in NTG patients showed that there is a significant difference in LGN volume between the control and advanced glaucoma group in the right and in the left side between the early and advanced glaucoma patients. Additionally, in early glaucoma the decrease in LGN volume was correlated with RNFL thinning measured with OCT. Similar to our previous results [25], such correlation was not observed for advanced glaucoma. Although the general tendency to decrease in LGN volume in more-advanced stages of glaucoma is observed, the relationships seem not to be very distinctive. There might be some reasons for this phenomenon. First, two main kinds of neurons in LGNs are described: interneurons confined to the LGN, and relay neurons with synapses to the visual cortex. Interneurons were suggested to be more resistant to transneuronal degeneration [32,36,37] during the glaucoma process. Similarly, two main types of pathways in LGN, magno- and parvocellular, are claimed to have different involvement during glaucoma. Moreover, the early glaucoma group may be the most heterogeneous, since before the VF changes occur the prominent RGC percentage needs to be injured. On the other side, our results may reflect the neurodegenerative process occurring independently in LGNs during NTG. The studies regarding neurodegenerative changes in central structures in the course of glaucoma are mainly conducted on animal models with high-tension glaucoma (HTG) and do not provide knowledge of the processes operating during NTG in humans. Advanced MRI techniques may help to obtain a clearer view regarding central involvement in human NTG.

The human cortex is a highly folded structure, so measuring cortical thickness based on MRI data requires complex reconstruction algorithms. The average thickness of the cerebral cortex for the whole brain is approximately 2.5 mm [38]. Obtaining MR images with submillimeter resolution is time-consuming and not always possible in clinical conditions. Typically, 3D T_1_-weighted images used for measurements are characterized by a resolution of 1.5 mm^3^ or less. To address these problems in our study, we used a technique called the surface-based registration (SBR) method that defines cortical thickness as the distance between two surfaces: grey/white and pial [39]. It allowed for precise calculation independent from voxel size and obtained results with accuracy around 10^−4^ m [39].

While analyzing the thickness of the cerebral cortex, we used three anatomical atlases [20,21,22,23], each one of them defining the areas of the visual cortex somewhat differently. The Destrieux atlas is the most detailed as compared to the Brodmann areas and Desikan and Killiany atlases. For example, only the Destrieux atlas contains an area called the occipital pole, while in the Desikan and Killiany atlas this region is included as a small part of a few neighboring labels. We aimed to check cortical abnormalities in NTG glaucoma in the most exact way possible, so we have decided to evaluate results according to different regional patterns.

The primary visual cortex is the predominant brain area receiving direct input related to visual stimulation. V1 is the first of the cortical regions to process information and is divided into six layers, each with different cell-types and functions. V1 responds to simple visual components such as orientation and direction [40]. V2 receives integrated information from V1, sends feedback connections to V1, and has feedforward connections with V3–V5. In our study, cortical thicknesses of V1 and V2 differed between the studied groups. The decrease (not significant) was observed in early glaucoma compared to control and, surprisingly, the increase in the cortical thickness was described in the advanced group. Similar increased volume of the components of the visual system, which enables for identification and classification of visual stimuli, was described by Williams [41]. The lack of statistical difference in V1 thickness between early glaucoma and control groups was reported previously for POAG [42], and was explained by the fact that the majority of retinal inputs project to layer 4 of the V1 area [43], receiving information from LGN. Hence, the possible selective degeneration of layer 4 in the V1 area may not significantly influence the whole V1 thickness [42].

V1 sends axons to the visual association areas [44]. Therefore, it could be speculated that degeneration of the visual pathway [45,46] could lead this region to increase its functional segregation ability in a compensatory capacity. According to our results, these compensatory mechanisms are observed in advanced, but not early glaucoma, which will support the hypothesis that the hypertrophy is the compensatory mechanism for visual-pathway atrophy rather than that temporary compensatory hypertrophy is followed by final atrophy. Our patients from the advanced group had bilaterally really advanced glaucoma with MD values near legal blindness. Increased cortical volume may be a sign of neuronal injury with cellular swelling or microglial activation [47,48]. Conversely, it may also reflect enhanced neuronal function or cortical plasticity [49,50]. There is growing evidence that extensive reorganization in response to sensory deprivation is possible in the adult brain and the anatomical changes may result from these functional adaptations. In the case of blindness, many studies have demonstrated that the visual association cortices are recruited for diverse functions including tactile, auditory, and linguistic tasks [51,52]. In this study, the tendencies similar to those observed for V1 were present also for cuneus and precuneus, areas classically related to visual information processing [53].

Fusiform gyrus and lateral occipital gyrus are located in the visual association cortex (V2), whose major function is the integration of visual information and generation of conscious perceptions [54]. It is said that V2 degenerates rapidly while V1 changes little during glaucoma progression. The reason is that V2 neurons may possibly be modulated by more-complex properties than V1, such as visual form [55,56] and motion processing [57,58], both of which are affected during POAG [42]. Additionally, V2 neurons are modulated by complex properties such as visual form and visual illusion [59], which may be impaired by the progression of the disease. Wang et al. [54] described that in HTG patients the V2 degenerated significantly while the primary visual cortex was relatively untouched. In NTG, we observed a significant increase in lateral occipital pole in advanced glaucoma in the left side, whereas there was no change in the right side. Similar lateralization to the left side of the damage was described previously [42,59,60,61,62], however, in our results some changes were more prominent for the right side. Our patients had a similar degree of glaucomatous damage in both eyes. Slightly different functions for each of the brain hemispheres were described, for example, face recognition [63], one of the tasks affected by glaucoma [64,65,66]. On the other hand, further studies on patients with similar stages of glaucoma bilaterally evaluating the influence of lateralization are needed to exclude bias.

To confirm the regions of the brain affected by glaucoma, FreeSurfer’s analysis was performed in QDEC, which revealed visual regions from the Desikan and Killiany anatomical atlas of altered cortical thickness. When compared to the control, the regions affected in early glaucoma with reduced thickness were the right lateral occipital gyrus and left lingual gyrus. These regions were also affected further during the course of glaucoma. The correlation between thinning of visual cortex and clinical measurements was previously described for glaucoma [42,59,61,67]. Correlation of gray matter volume with glaucoma severity was also demonstrated by significant differences in visual cortex thickness between mild and severe glaucoma groups, with increased damage in late-stage glaucoma [59,68]. In our study, when comparing early and advanced groups the most prominent difference was an increase in the GM thickness in the right cuneus during the course of glaucoma. One of the cuneus functions seems to be to integrate the somatosensory information with other sensory stimuli and cognitive processes such as attention, learning, and memory [53]. These results indicate that NTG predominantly involves visual cortices, accompanied by abnormalities of other cortices associated with attention and cognition. However, the interactions between these cortices remain unclear. NTG changes the information processing in the brain by reorganizing the information flows between visual cortices and non-visual brain areas, and these alterations are consistent with microstructural injury of visual cortices [69]. Li reported decreased information outflows in NTG patients from the left BA17 to bilateral cuneus and increased effective connectivity, reflecting integration within a distributed system, from BA19 and BA18 to cuneus (BA17) [70].

When evaluating patients with glaucoma, the structure (RNFL) and function (VF) of the visual pathway are measured and, usually, they are not strictly related. The assessment of early and late stages of the disease is particularly challenging with only structural defects in early phase and functional progression accompanied by stable RNFL thickness in advanced stages. The interesting finding in our study is the negative correlation between MD value resembling the depth of functional deficit and the thickness of BA17 observed in advanced glaucoma, which may support the hypothesis that in advanced glaucoma the measurement of visual cortex thickness may be an additional diagnostic tool to assess possible progression. Qing et al. [71] also showed that POAG neuropathy leads to decreased cortical activity in the primary visual cortex corresponding to the central normal visual field, but not OCT measurements and ophthalmoscopy. In no glaucoma stage were the visual cortices measurements correlated with RNFL thickness. Such correlations were present only for early glaucoma and LGN volume, the first cortical visual center.

## 5. Conclusions

To sum up, in this study we evaluated changes in the brain regions connected with the visual functions in different NTG stages. We showed the increase in the GM thickness in the V1 region with more-advanced glaucoma stages correlated with VF parameters, which may show compensatory hypertrophy. Additionally, the regions of the brain affected early during glaucoma with reduced thickness were the right lateral occipital gyrus and left lingual gyrus. When comparing early and advanced groups, the most prominent difference was increase in the GM thickness in the right cuneus during the course of glaucoma.

## Figures and Tables

**Figure 1 jcm-11-01907-f001:**
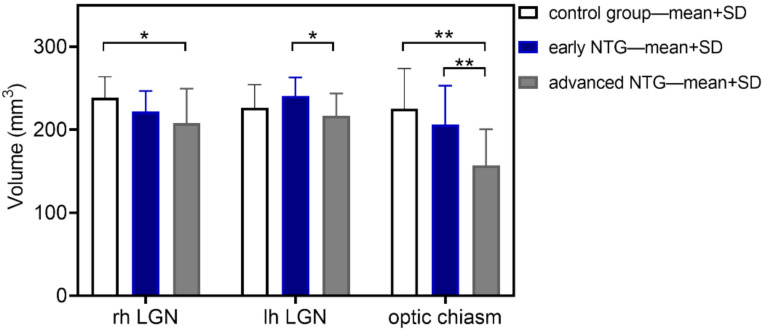
Results of voxel-based morphometry analysis of LGN and optic chiasm volume in studied groups: control (white), early (blue), and advanced (grey) NTG (* *p* < 0.05, ** *p* < 0.01).

**Figure 2 jcm-11-01907-f002:**
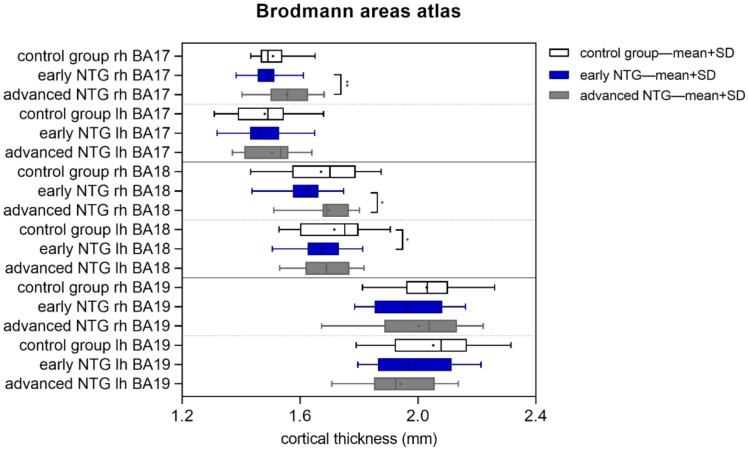
Cortical thickness analysis results in the control group and NTG patients (in early and advanced stage) measured according to the Brodmann area atlas in primary visual cortex (V1–BA17), secondary visual cortex (V2–BA18), and middle temporal visual area (V5/MT–BA19) (* *p* < 0.05, ** *p* < 0.01).

**Figure 3 jcm-11-01907-f003:**
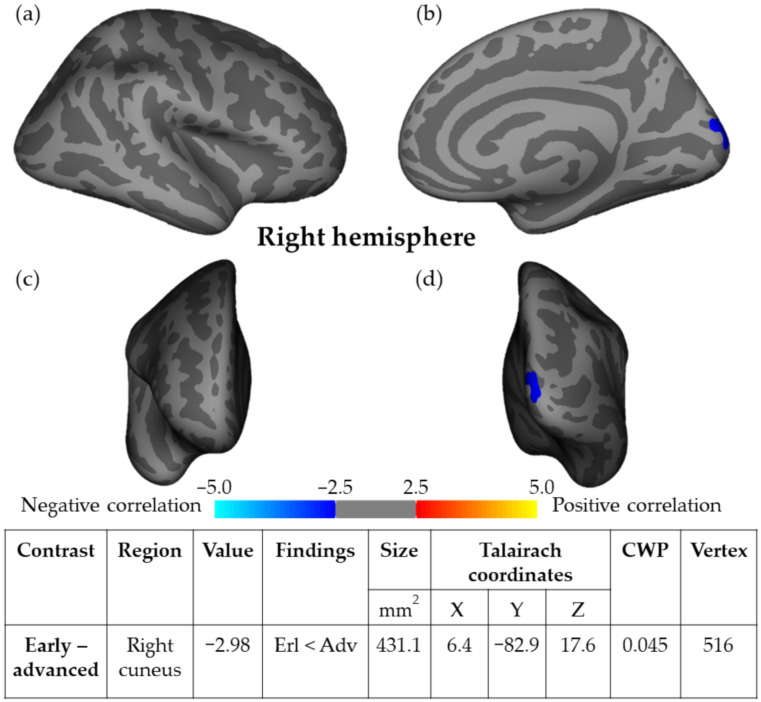
Cortical thickness maps showing significant thinning in the right hemisphere cuneus in early-stage NTG as compared to advanced NTG stage in (**a**) lateral, (**b**) medial, (**c**) anterior, and (**d**) posterior views. Results were obtained using Monte Carlo simulation, with a threshold of *p* < 0.01, to provide clusterwise correction for multiple comparisons.

**Table 1 jcm-11-01907-t001:** Demographic and clinical characteristics of the studied groups.

	Control Group *n* = 17	Early Glaucoma *n* = 28	Advanced Glaucoma *n* = 17
Age	68.5 ± 11.8	63.2 ± 11.4	65 ± 14.7
Male/female	4/13	8/20	5/11
Mean visual field MD (dB)—right eye	-	−2.97 (from −5.88 to −0.11)	−19.45 (form −29.53 to −16.18)
Mean visual field MD (dB)—left eye	-	−3.96 (from −5.93 to −0.43)	−18.33 (form −25.48 to −14.1)

**Table 2 jcm-11-01907-t002:** Mean cortical thickness results in selected visual regions based on Destrieux cortical parcellation atlas in control and NTG groups. Statistically significant results are marked bold.

Destrieux Anatomical Atlas								
Anatomical Structure Name	Side	Control Group	Early-Stage NTG	Advanced-Stage NTG	ANOVA	ANOVA Post-Hoc (*p*-Value)		
						Con vs. Erl	Con vs. Adv	Erl vs. Adv
Lingual gyrus	right	1.51	1.49	1.54	0.131	0.795	0.427	0.110
	left	1.65	1.61	1.63	0.478	0.446	0.794	0.873
	bilateral	1.58	1.55	1.58	0.301	0.520	0.949	0.325
Calcarine sulcus	right	2.10	2.09	2.02	0.147	0.941	0.181	0.219
	left	1.71	1.71	1.76	0.334	1.000	0.424	0.356
	bilateral	1.89	1.90	1.89	0.988	0.987	0.995	0.999
Cuneus	right	1.61	1.56	1.65	**0.016**	0.206	0.541	**0.013**
	left	1.65	1.61	1.63	0.478	0.446	0.794	0.873
	bilateral	1.63	1.58	1.64	**0.024**	0.105	0.909	**0.034**
Collateral sulcus and lingual sulcus	right	2.01	2.04	2.01	0.758	0.863	0.988	0.767
	left	1.69	1.63	1.71	**0.032**	0.188	0.757	**0.034**
	bilateral	1.87	1.86	1.84	0.823	0.934	0.971	0.814
Occipital pole	right	1.51	1.50	1.49	0.905	0.918	0.918	0.999
	left	1.67	1.68	1.78	**0.007**	0.966	**0.014**	**0.013**
	bilateral	1.59	1.59	1.64	0.247	0.989	0.382	0.248

NTG—normal-tension glaucoma, Con—control group, Erl—early-stage NTG, Adv—advanced-stage NTG.

**Table 3 jcm-11-01907-t003:** Cortical regions from Desikan and Killiany atlas displaying change in thickness between groups—results before clusterwise correction for multiple comparisons.

Contrast	Anatomical Region—Desikan and Killiany Atlas	Value	Findings	Size	Talairach Coordinates			Vertex
				mm^2^	X	Y	Z	
Control–early	Right lateral occipital gyrus	2.98	Con > Erl	101.3	39.1	−81.8	2.5	144
	Left lingual gyrus	2.06	Con > Erl	5.2	−3.9	−87.8	−6.2	6
Control–advanced	Right lateral occipital gyrus	3.10	Con > Adv	103.8	45.6	−79.9	0.9	150
	Right pericalcarine	−2.99	Con < Adv	97.4	13.6	−91.7	7.1	131
	Left lingual gyrus	2.62	Con > Adv	25.4	−22.1	−54.5	−1.7	75
	Left lingual gyrus	2.06	Con > Adv	5.5	−15.8	−59.1	−5.3	9
Early–advanced	Right lateral occipital gyrus	3.08	Erl > Adv	98.6	44.7	−78.2	0.2	137
	Right cuneus	−2.97	Erl < Adv	431.1	6.5	−86.2	14.6	516
	Right lingual gyrus	−2.84	Erl < Adv	237.6	4.5	−85.7	−5	257
	Right lateral occipital gyrus	−2.13	Erl < Adv	12.6	21.4	−92.8	−8.5	13
	Left lateral occipital gyrus	2.46	Erl > Adv	41.8	−40	−85	−6.5	59

NTG—normal-tension glaucoma, Con—control group, Erl—early-stage NTG, Adv—advanced-stage NTG.

**Table 4 jcm-11-01907-t004:** Parametric (Pearson’s) and nonparametric (Spearman’s) correlation analyses between LGN volume and the clinical parameters (MD and RNFL) in studied groups. Statistically significant results are marked bold.

	Pearson Correlation						Spearman Correlation					
	Control		Early NTG		Advanced NTG		Control		Early NTG		Advanced NTG	
	*r*	*p*-Value	*r*	*p*-Value	*r*	*p*-Value	*R*	*p*-Value	*R*	*p*-Value	*R*	*p*-Value
MD and RNFL vs. mean LGN volume—corresponding eyes												
MD	-	-	−0.13	0.424	−0.05	0.797	-	-	−0.12	0.465	−0.04	0.864
Average RNFL	−0.05	0.859	0.16	0.317	−0.02	0.926	−0.14	0.625	0.16	0.315	0.02	0.945
Superior RNFL	−0.33	0.251	0.16	0.321	0.33	0.186	−0.15	0.599	0.14	0.381	0.20	0.433
Temporal RNFL	0.09	0.756	**0.34**	**0.027**	0.03	0.915	0.16	0.594	0.29	0.059	0.18	0.486
Inferior RNFL	0.11	0.703	0.07	0.659	−0.13	0.593	0.35	0.226	0.02	0.908	−0.07	0.798
Nasal RNFL	0.20	0.498	−0.14	0.368	−0.37	0.130	0.04	0.887	−0.14	0.373	**−0.50**	**0.035**
MD and RNFL vs. mean LGN volume—opposite eyes												
MD	-	-	0.12	0.450	0.18	0.393	-	-	0.15	0.359	0.13	0.521
Average RNFL	0.15	0.597	**0.38**	**0.012**	0.08	0.758	0.07	0.805	**0.47**	**0.002**	0.31	0.206
Superior RNFL	−0.06	0.844	0.26	0.103	0.41	0.088	0.12	0.675	**0.43**	**0.005**	0.38	0.122
Temporal RNFL	0.48	0.082	**0.39**	**0.011**	0.30	0.227	0.46	0.094	**0.50**	**0.001**	0.44	0.067
Inferior RNFL	0.18	0.548	**0.37**	**0.016**	−0.15	0.553	0.18	0.532	**0.36**	**0.019**	−0.07	0.773
Nasal RNFL	0.09	0.762	0.04	0.805	**−0.48**	**0.043**	0.17	0.557	0.07	0.670	**−0.51**	**0.029**

**Table 5 jcm-11-01907-t005:** Parametric (Pearson’s) and nonparametric (Spearman’s) correlation analyses between mean cortical thickness in selected areas and the clinical parameters (MD and RNFL) in studied groups. Statistically significant results are marked bold.

	Pearson Correlations						Spearman Correlations					
	Control		Early NTG		Advanced NTG		Control		Early NTG		Advanced NTG	
	*r*	*p*-Value	*r*	*p*-Value	*r*	*p*-Value	*R*	*p*-Value	*R*	*p*-value	*R*	*p*-Value
MD vs. mean cortical thickness—corresponding eyes												
V1–BA17			−0.08	0.608	**−0.50**	**0.009**			−0.12	0.480	**−0.52**	**0.006**
V2–BA18			−0.25	0.119	−0.32	0.121			−0.25	0.123	−0.34	0.105
V5/MT–BA19			0.02	0.871	−0.38	0.064			0.08	0.576	**−0.50**	**0.011**
Lingual gyrus			−0.04	0.781	−0.06	0.777			−0.02	0.906	−0.20	0.316
Calcarine sulcus			0.30	0.058	−0.13	0.551			**0.33**	**0.034**	−0.05	0.807
Cuneus			−0.27	0.079	−0.16	0.443			−0.23	0.148	−0.18	0.366
Collateral sulcus and lingual sulcus			0.17	0.281	−0.17	0.413			0.20	0.202	−0.08	0.693
Occipital pole			**−0.30**	**0.048**	0.08	0.717			**−0.35**	**0.019**	−0.06	0.773
MD vs. mean cortical thickness—opposite eyes												
V1–BA17			−0.24	0.143	**−0.40**	**0.046**			−0.29	0.072	**−0.48**	**0.014**
V2–BA18			0.03	0.857	−0.16	0.466			0.01	0.942	−0.19	0.375
V5/MT–BA19			0.04	0.799	−0.10	0.619			0.03	0.834	−0.17	0.410
Lingual gyrus			0.07	0.634	−0.28	0.170			0.11	0.475	−0.28	0.180
Calcarine sulcus			−0.24	0.128	0.11	0.605			−0.24	0.126	−0.04	0.855
Cuneus			0.07	0.640	−0.13	0.521			0.10	0.512	−0.19	0.365
Collateral sulcus and lingual sulcus			−0.13	0.399	0.26	0.218			−0.12	0.459	0.09	0.689
Occipital pole			0.04	0.796	−0.15	0.485			0.05	0.763	−0.09	0.658
Average RNFL vs. mean cortical thickness—corresponding eyes												
V1–BA17	−0.15	0.591	0.00	0.980	−0.08	0.750	−0.13	0.634	0.13	0.382	−0.10	0.685
V2–BA18	−0.40	0.135	−0.14	0.345	−0.31	0.230	−0.32	0.240	−0.08	0.617	−0.14	0.582
V5/MT–BA19	−0.45	0.127	0.05	0.756	−0.41	0.087	−0.19	0.540	0.12	0.389	−0.37	0.130
Lingual gyrus	−0.27	0.314	0.11	0.458	0.12	0.668	−0.24	0.374	0.21	0.171	0.13	0.642
Calcarine sulcus	0.18	0.520	0.10	0.524	−0.22	0.406	0.09	0.746	0.10	0.494	−0.23	0.367
Cuneus	−0.31	0.240	0.00	0.977	0.20	0.432	−0.28	0.299	0.05	0.757	0.15	0.574
Collateral sulcus and lingual sulcus	0.13	0.630	0.08	0.603	−0.21	0.412	0.00	0.987	0.06	0.661	−0.30	0.229
Occipital pole	−0.19	0.476	−0.09	0.532	0.06	0.828	−0.36	0.175	−0.05	0.756	0.08	0.761
Average RNFL vs. mean cortical thickness—opposite eyes												
V1–BA17	−0.16	0.578	−0.17	0.273	0.08	0.755	−0.16	0.580	−0.07	0.664	0.04	0.880
V2–BA18	−0.36	0.188	0.02	0.885	−0.14	0.599	−0.22	0.440	0.04	0.767	−0.01	0.957
V5/MT–BA19	−0.25	0.401	0.13	0.361	−0.12	0.629	−0.12	0.703	0.20	0.155	−0.06	0.812
Lingual gyrus	−0.14	0.605	0.21	0.167	−0.36	0.166	−0.25	0.356	0.19	0.206	−0.41	0.114
Calcarine sulcus	0.07	0.808	0.01	0.960	−0.04	0.892	−0.01	0.965	0.00	0.982	0.15	0.568
Cuneus	−0.27	0.307	−0.01	0.940	0.00	0.987	−0.25	0.356	0.06	0.682	0.07	0.796
Collateral sulcus and lingual sulcus	−0.08	0.779	−0.03	0.859	0.05	0.839	−0.06	0.833	−0.03	0.857	0.33	0.182
Occipital pole	0.12	0.654	0.09	0.554	−0.28	0.294	0.10	0.724	0.14	0.339	−0.33	0.218

## Data Availability

The data that support the findings of this study are available from the corresponding author upon reasonable request.

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
