# Peer review of "Visual Tract Degradation in Bilateral Normal-Tension Glaucoma—Cortical Thickness Maps and Volumetric Study of Visual Pathway Areas"

_jcm, 2022, doi:10.3390/jcm11071907_

Round 1
Reviewer 1 Report
Authors have done interesting work to evaluate the changes in the visual pathways like Optic chiasma, lateral geniculate body, or cortical thickness in the occipital region using MRI, in patients with Normal-tension glaucoma (NTG) and have found significant correlations.
Following are the comments related to study
- The criteria for diagnosis of NTG does not involve full-day diurnal variation (DVT), not even office DVT, that may cause an overlap in POAG, and also pachymetry is not taken into consideration.
- Authors have included patients with the bilaterally same stage of defect, which is usually not the case in NTG, however, authors have not specified the period over which they would have collected 45 patients.
- The methodology section involving MRI details has to be evaluated by a radiologist. Further, the algorithms and software used to calculate thicknesses in different areas and to measure the size of the lateral geniculate body or different areas of the brain have to be standardized and evaluated by the radiologist.
- The authors have found significant differences between normal and advanced glaucoma in almost all parameters, however, there are no comparisons with other forms of glaucoma. It is possible that such changes will occur in other forms of glaucoma as well; hence, a comparison with other types of glaucoma would establish the association better. Only if it is established that these changes are related to NTG and not in other types of glaucoma, then this may work as a tool to look at the cause of damage in NTG.
- Authors in the discussion have mentioned that the changes in the visual cortex in advanced glaucoma may help in detecting progression; however, that seems to be extrapolation beyond data. The data does not indicate whether the changes seen in the brain in relation to glaucoma are cause or effect. Further, whether they change with the progression of the disease is another research question that would involve longitudinal studies. In addition, one may not require MRI to detect glaucoma, as it would be very evident in simple optic nerve evaluation and visual fields.
Author Response
Response to Reviewer 1
Authors have done interesting work to evaluate the changes in the visual pathways like Optic chiasma, lateral geniculate body, or cortical thickness in the occipital region using MRI, in patients with Normal-tension glaucoma (NTG) and have found significant correlations.
RESPONSE: The authors would like to thank the Reviewer for all the valuable comments.
Following are the comments related to study
- The criteria for diagnosis of NTG does not involve full-day diurnal variation (DVT), not even office DVT, that may cause an overlap in POAG, and also pachymetry is not taken into consideration.
RESPONSE 1: The diagnosis of NTG was made according to IOP measurements made at diagnosis during the office hours. The IOP assessment was made at 8, 10,12, and 18 o’clock . NTG was diagnosed when the measurement never exceeded 21 mmHg. The daily curve are nowadays not frequently used because they are time consuming and there are doubts concerning the night measurements and the IOP fluctuations connected with body position change [1]. When the patients were diagnosed, IOP was corrected according to the CCT. However, according to the European Glaucoma Guideline published last year CCT corrected measurements of IOP are listed as the things that should be avoided [2]. Before the inclusion to the study, the authors rechecked the basic IOP measurements to ensure that also raw IOP data enabled to include patients as NTG subjects. None of the patients was excluded as the CCT were normal or lower, never increased.
- Ho CH, Wong JKW. Role of 24-Hour Intraocular Pressure Monitoring in Glaucoma Management. J Ophthalmol. 2019 Sep 19;2019:3632197. doi: 10.1155/2019/3632197. PMID: 31641532; PMCID: PMC6770303.
- European Glaucoma Society. Terminology and Guidelines for Glaucoma. 2020
- Authors have included patients with the bilaterally same stage of defect, which is usually not the case in NTG, however, authors have not specified the period over which they would have collected 45 patients.
RESPONSE 2: The ophthalmologists who are the co-authors of the manuscript work in the Glaucoma Department and take care of more than 2500 glaucoma patients, approximately 500 of whom are diagnosed as NTG in different stages. Our group of NTG patients was characterized in some publications [1]. The authors agree with the Reviewer that NTG especially in early stages tends to be asymmetric, one of our manuscripts examines such group of patients [2]. However, our established group of patients enables as to find the patients fitting the inclusion criteria.
- Kosior-Jarecka E, Wróbel-Dudzińska D, Łukasik U, Żarnowski T. Disc haemorrhages in Polish Caucasian patients with normal tension glaucoma. Acta Ophthalmol. 2019 Feb;97(1):68-73. doi: 10.1111/aos.13848. Epub 2018 Oct 3. PMID: 30284408.
- Kosior-Jarecka E, Wróbel-Dudzińska D, Pietura R, Pankowska A, Szczuka B, Żarnowska I, Łukasik U, Żarnowski T. Results of Neuroimaging in Patients with Atypical Normal-Tension Glaucoma. Biomed Res Int. 2020 Aug 18;2020:9093206. doi: 10.1155/2020/9093206. PMID: 32908924; PMCID: PMC7450348.
- The methodology section involving MRI details has to be evaluated by a radiologist. Further, the algorithms and software used to calculate thicknesses in different areas and to measure the size of the lateral geniculate body or different areas of the brain have to be standardized and evaluated by the radiologist.
RESPONSE 3: The manuscript was collectively prepared by ophthalmologists and radiologists. The ophthalmic examination was performed by ophthalmologists. The MRI scanning and its evaluation were made by radiologists. The information was added in the Material and method section pointing the name of the authors responsible for the tasks.
- The authors have found significant differences between normal and advanced glaucoma in almost all parameters, however, there are no comparisons with other forms of glaucoma. It is possible that such changes will occur in other forms of glaucoma as well; hence, a comparison with other types of glaucoma would establish the association better. Only if it is established that these changes are related to NTG and not in other types of glaucoma, then this may work as a tool to look at the cause of damage in NTG.
RESPONSE 4: The authors believe that NTG is a separate form of glaucoma with the causative factors different than elevated IOP. One of the proposed mechanisms is the neurodegeneration in central nervous system. That is why our interest is focused on this group. Additionally, in the literature there are studies concerning volumetry of cortical thickness in unspecified groups of glaucoma patients or POAG [1-3]. However, there is scant of information about NTG patients especially fulfilling so strict inclusion criteria as in our study. The results of this study inspired us to design the research comparing brain changes in NTG and HTG patients which are now ongoing.
- Fukuda M, Omodaka K, Tatewaki Y, Himori N, Matsudaira I, Nishiguchi KM, Murata T, Taki Y, Nakazawa T. Quantitative MRI evaluation of glaucomatous changes in the visual pathway. PLoS One. 2018 Jul 9;13(7):e0197027. doi: 10.1371/journal.pone.0197027. PMID: 29985921; PMCID: PMC6037347.
- Chen WW, Wang N, Cai S, Fang Z, Yu M, Wu Q, Tang L, Guo B, Feng Y, Jonas JB, Chen X, Liu X, Gong Q. Structural brain abnormalities in patients with primary open-angle glaucoma: a study with 3T MR imaging. Invest Ophthalmol Vis Sci. 2013 Jan 17;54(1):545-54. doi: 10.1167/iovs.12-9893. PMID: 23258150.
- Li C, Cai P, Shi L, Lin Y, Zhang J, Liu S, Xie B, Shi Y, Yang H, Li S, Du H, Wang J. Voxel-based morphometry of the visual-related cortex in primary open angle glaucoma. Curr Eye Res. 2012 Sep;37(9):794-802. doi: 10.3109/02713683.2012.683506. Epub 2012 May 25. PMID: 22631870.
- Authors in the discussion have mentioned that the changes in the visual cortex in advanced glaucoma may help in detecting progression; however, that seems to be extrapolation beyond data. The data does not indicate whether the changes seen in the brain in relation to glaucoma are cause or effect. Further, whether they change with the progression of the disease is another research question that would involve longitudinal studies. In addition, one may not require MRI to detect glaucoma, as it would be very evident in simple optic nerve evaluation and visual fields.
RESPONSE 5: The authors believe that the changes in lateral geniculate body during the course of glaucoma are a hypothetic target to evaluate the progression of glaucoma in advanced cases. Nowadays the detection of progression in this group is difficult with no effective strategies in VF (static strategies which are designed only for sign size 3, but not 5) and OCT (in advanced cases RNFL measurements did not decrease below 50mm because so called ”floor effect”). The LGN consists mainly of the retinal ganglion cell axons which possibly could enable for the observation of change in measurements when the methods traditionally used in glaucoma practice are disappointing and inadequate. The authors carry ongoing study on 7T MRI focused on this task, which is the continuation of our previous results [1]. Of course, MRI is not a tool to be used in everyday practice to diagnose glaucoma with the exception of ”atypical” forms of NTG.
- Kosior-Jarecka E, Pankowska A, Polit P, Stępniewski A, Symms MR, Kozioł P, Żarnowski T, Pietura R. Volume of Lateral Geniculate Nucleus in Patients with Glaucoma in 7Tesla MRI. J Clin Med. 2020 Jul 26;9(8):2382. doi: 10.3390/jcm9082382. PMID: 32722571; PMCID: PMC7466157.
Reviewer 2 Report
The present study evaluated the changes in the central visual pathways during early and advanced stages in bilateral normal-tension glaucoma (NTG). The authors concluded that increase in the grey matter thickness in the V1 region with more advanced glaucoma stages may reflect compensatory hypertrophy. Furthermore, when comparing early and advanced groups the most prominent difference was increase of the grey matter thickness in the right cuneus.
As a general comment is an interesting study. I believe however that the distinction between early and advanced glaucoma is confusing. For example, the authors reported that significant difference in lateral geniculate nuclei volume between the control and advanced glaucoma group in the right hemisphere (p=0.03) and in the left hemisphere between the early and advanced glaucoma patients (p=0.026). It is surprising that there was a difference between early and advanced and not between control and advanced
Α Minor comment
Methods:
In visual field examination, patients were divided into two sub- 113 groups: early glaucoma (MD less than -6 dB; n=28) and advanced glaucoma (MD less than 114 -12 dB; Needs correction
Author Response
Response to Reviewer 2
The present study evaluated the changes in the central visual pathways during early and advanced stages in bilateral normal-tension glaucoma (NTG). The authors concluded that increase in the grey matter thickness in the V1 region with more advanced glaucoma stages may reflect compensatory hypertrophy. Furthermore, when comparing early and advanced groups the most prominent difference was increase of the grey matter thickness in the right cuneus.
As a general comment is an interesting study. I believe however that the distinction between early and advanced glaucoma is confusing. For example, the authors reported that significant difference in lateral geniculate nuclei volume between the control and advanced glaucoma group in the right hemisphere (p=0.03) and in the left hemisphere between the early and advanced glaucoma patients (p=0.026). It is surprising that there was a difference between early and advanced and not between control and advanced
RESPONSE: The authors would like to thank the Reviewer for all the valuable comments.
The classification of early and advanced glaucoma was performed according to broadly used criteria of Parish-Anderson- Hodapp focusing on changes in visual field. The correct inclusion and division of the patients in this study is confirmed by the measurements of anterior visual pathway in which up to LGN we observed the statistically significant decrease between early and advanced group. The authors believe that the PHA criteria and staging of glaucoma according to VF changes is not perfect (knowing that about 50% of RGC dies before the changes in VF appear). However, nowadays there is no better criteria to assess the staging.
The difference between early and advanced but not control cases in cortex thickness in some regions is the main, but surprising, finding of our study. It shows that at the beginning of glaucoma (at least NTG which was examined in our study), the cortex thickness diminishes as the results of apoptosis of RNFL but later the “arising holes” are fulfilled with the tissue causing hyperthropy which results in the increase of the cortical thickness, however, the nature of the fulfilling tissue was not aimed to be identified in this study.
Α Minor comment
Methods:
In visual field examination, patients were divided into two sub- 113 groups: early glaucoma (MD less than -6 dB; n=28) and advanced glaucoma (MD less than 114 -12 dB; Needs correction
RESPONSE: The sentence was corrected as follows: “According to the criteria of Parish-Hoddap-Anderson in visual field examination, patients were divided into two subgroups: early glaucoma (MD better than -6 dB; n=28) and advanced glaucoma (MD less than -12 dB; n=17).“
Round 2
Reviewer 2 Report
Accept as it is